# Electrical synapse structure requires distinct isoforms of a postsynaptic scaffold

**Jennifer Carlisle Michel**, **Margaret M. B. Grivette**, **Amber T. Harshfield**, **Lisa Huynh**, **Ava P. Komons**, **Bradley Loomis**, **Kaitlan McKinnis**, **Brennen T. Miller**, **Ethan Q. Nguyen**, **Tiffany W. Huang**, **Sophia Lauf**, **Elias S. Michel**, **Mia E. Michel**, **Jane S. Kissinger**, **Audrey J. Marsh**, **William E. Crow**, **Lila E. Kaye**, **Abagael M. Lasseigne**, **Rachel M. Lukowicz-Bedford**, **Dylan R. Farnsworth**, **E. Anne Martin**, **Adam C. Miller**\*

Institute of Neuroscience, Department of Biology, University of Oregon, Eugene, Oregon, United States of America

\* acmiller@uoregon.edu

**Data Availability Statement:** All relevant data are in the manuscript and its supporting information files. Animal lines are available from the Zebrafish

## Abstract

Electrical synapses are neuronal gap junction (GJ) channels associated with a macromolecular complex called the electrical synapse density (ESD), which regulates development and dynamically modifies electrical transmission. However, the proteomic makeup and molecular mechanisms utilized by the ESD that direct electrical synapse formation are not well understood. Using the Mauthner cell of zebrafish as a model, we previously found that the intracellular scaffolding protein ZO1b is a member of the ESD, localizing postsynaptically, where it is required for GJ channel localization, electrical communication, neural network function, and behavior. Here, we show that the complexity of the ESD is further diversified by the genomic structure of the ZO1b gene locus. The ZO1b gene is alternatively initiated at three transcriptional start sites resulting in isoforms with unique N-termini that we call ZO1b-Alpha, -Beta, and -Gamma. We demonstrate that ZO1b-Beta and ZO1b-Gamma are broadly expressed throughout the nervous system and localize to electrical synapses. By contrast, ZO1b-Alpha is expressed mainly non-neuronally and is not found at synapses. We generate mutants in all individual isoforms, as well as double mutant combinations *in cis* on individual chromosomes, and find that ZO1b-Beta is necessary and sufficient for robust GJ channel localization. ZO1b-Gamma, despite its localization to the synapse, plays an auxiliary role in channel localization. This study expands the notion of molecular complexity at the ESD, revealing that an individual genomic locus can contribute distinct isoforms to the macromolecular complex at electrical synapses. Further, independent scaffold isoforms have differential contributions to developmental assembly of the interneuronal GJ channels. We propose that ESD molecular complexity arises both from the diversity of unique genes and from distinct isoforms encoded by single genes. Overall, ESD proteomic diversity is expected to have critical impacts on the development, structure, function, and plasticity of electrical transmission.

International Resource Center (ZIRC, https://zebrafish.org/fish/lineAll.php).

**Funding:** This work was supported by the NIH Eunice Kennedy Shriver National Institute of Child Health and Human Development (NICHD) Developmental Biology Training Grant T32HD007348 to LEK and AML, the NIH National Institute of General Medical Sciences (NIGMS) Graduate Training in Genetics Grant T32GM007413 to RML and WEC, the NICHD Ruth L Kirschstein National Research Service Award F32HD102182 to EAM, the Hui Women in Science and Math (WiSM) Academic Residential Community (ARC) Summer Research Award to APK, and the National Institute of Neurological Disorders and Stroke (NINDS) Awards R21NS117967 and R01NS105758 to ACM. The funders had no role in study design, data collection and analysis, decision to publish, or preparation of the manuscript.

**Competing interests:** The authors have declared that no competing interests exist.

## Author summary

Brain function, including thought, emotion, and action, relies on the patterns and properties of connections amongst neurons. The connections, called synapses, come in two main types, electrical and chemical. Appropriate formation during development is critical, as disruptions to electrical and chemical synapse assembly can result in disorders such as autism and epilepsy. While chemical synapses are well studied, little is known about the mechanisms of electrical synapse formation.

Electrical synapses are formed by gap junction (GJ) channels between neurons, providing direct, fast communication. We previously found that the scaffolding molecule ZO1 is required for GJ localization, synaptic function, and for an escape response that allows animals to avoid predation. Here, we show that the ZO1 gene is complex, encoding multiple distinct proteins (isoforms). Distinct, independent ZO1 isoforms localize to electrical synapses, yet each contributes differentially to assembly. Thus, like chemical synapses, electrical synapse assembly is regulated by associated proteins that are diverse, even when viewed from the perspective of only a single gene, and that are necessary for neurotypical function.

Ultimately, we expect electrical synapses protein complexity to include distinct isoforms from many diverse genes, that together will shape the development, function, and plasticity of electrical transmission.

## Introduction

Fast synaptic transmission in the nervous system is mediated by electrical and chemical synapses, with each type imparting distinct modes of communication [1–3]. Electrical synapses directly couple neurons via interneuronal gap junction (GJ) channels, whereas chemical synapses utilize neurotransmitter release/reception for signaling. Despite unique structures and functions, both synaptic types are supported by locally-assembled, cytoplasmic, associated proteins that regulate the development and function of transmission. At chemical synapses, hundreds of proteins are associated with the presynaptic active zone (AZ) and postsynaptic density (PSD) and regulate synaptic vesicle release and neurotransmitter receptor localization, respectively [4–7]. Critical to the structure and function of glutamatergic chemical synapses is a family of postsynaptic membrane-associated guanylate kinases (MAGUKs) that coordinate and link the macromolecular construction of the cytoplasmic PSD structure to the membrane localization and function of NMDA and AMPA receptors [8, 9]. By contrast, few electrical synapse-associated proteins have been identified to date, yet electron micrographs of neuronal tissue suggest a cytoplasmic, GJ-associated regulatory network complementing each side of electrical synapses [10, 11], and we have termed these structures the electrical synapse density (ESD). Our recent work revealed that the ESD component ZO1b, a MAGUK intracellular scaffolding protein, localizes postsynaptically at zebrafish electrical synapses where it directly interacts with neuronal GJ-channel forming Connexins, and is required for GJ-channel localization, electrical communication, neural network function, and behavior [12]. Yet the proteomic diversity found at the ESD and how it controls electrical synapse formation remains poorly understood.

Synaptic proteomic diversity can arise from both unique gene products as well as distinct isoforms of individual genes. At glutamatergic chemical synapses, the alternative transcriptional initiation of postsynaptic MAGUKs creates distinct protein isoforms with unique N-

terminal domains that result in differential regulation and activity. For example, the MAGUK protein SAP97 exists as two alternatively initiated isoforms, α and β. The SAP97α-isoform is palmitoylated on its unique N-terminal domain, targeting the isoform to the PSD. The SAP97β-isoform encodes an L27 interaction domain that targets this isoform to the non-PSD, perisynaptic region. Expression of these different SAP97 isoforms allows distinct classes of associated complexes to control the subsynaptic distribution of AMPA receptors, consequently affecting synaptic strength [13, 14]. By contrast, there are no reports revealing similar isoform-specific contributions to ESD proteomic diversity, yet the ESD-associated ZO1 is better known for its role in regulating epithelial tight junction (TJ) structure and function [15]. In epithelia, two isoforms of ZO1 arise from alternative splicing, are associated with anatomically distinct TJs, and cause isoform-dependent effects on junctional plasticity that alter epithelial barrier function [16, 17]. These observations suggest that unique isoforms of ESD proteins may have distinct spatiotemporal and functional motifs that control the structure and function of electrical synapses.

Here we identify an alternative initiation mechanism of ZO1b that regulates electrical synapse diversity and development. We investigate three independent transcriptional start sites of the ZO1b-encoding gene, hereafter called ZO1b-Alpha, -Beta, and -Gamma, and use genetics, transgenics, and imaging to elucidate their contributions and functions in the Mauthner cell neural circuit of zebrafish. The results expand the proteomic complexity of the ESD and reveal that unique isoforms encoded from a single gene can have differential effects on the structure and function of electrical synapse formation. This emerging complexity of ESD proteins and functions at the electrical synapse parallels findings at chemical synapses where unique scaffold isoforms play distinct functions in the structure, function, and plasticity of synaptic transmission.

## Results

### Alternatively initiated isoforms of the *tjp1b/ZO1b gene* are expressed in the Mauthner cell circuit

At electrical synapses, interneuronal gap junction (GJ) channels are formed by the docking of two hemichannels, composed of Connexin proteins in vertebrates [18]. Hemichannels are contributed by each of the coupled neurons to create the interconnecting channel, forming a direct path of communication that allows the spread of electrical currents and small metabolites between neurons. Our recent work demonstrated that the ZO1b intracellular scaffold is required postsynaptically for Connexin localization at zebrafish Mauthner cell electrical synapses (Fig 1A) [12, 19]. Located on chromosome 25, the *tjp1b* gene encodes the ZO1b protein (*tjp1b/ZO1b*), and the locus is complex with three distinct initiation sites annotated on public databases (Ensembl, ENSDARG00000079374) that predict three transcripts of varying lengths (Fig 1B). Each transcript contains a unique first exon, including distinct 5' untranslated regions (UTRs) and translational start codons, and each has a varying number of additional exons incorporated before splicing into a common exon and the subsequent remaining transcript and 3' UTR (Fig 1C). While each transcript predicts a unique N-terminal peptide sequence, all three retain the assortment of common MAGUK protein-protein interaction modules including PDZs, SH3, GUK, and ZU5 domains (Fig 1D). This genomic structure spurred us to examine if *tjp1b/ZO1b* transcriptional initiation complexity contributes to electrical synapse proteomic diversity.

To examine how alternatively initiated transcripts might contribute to Mauthner electrical synapses, we first inspected the spatiotemporal expression pattern of *tjp1b/ZO1b* through early development using our single cell RNA-seq (scRNA-seq) atlas [20, 21]. The atlas revealed

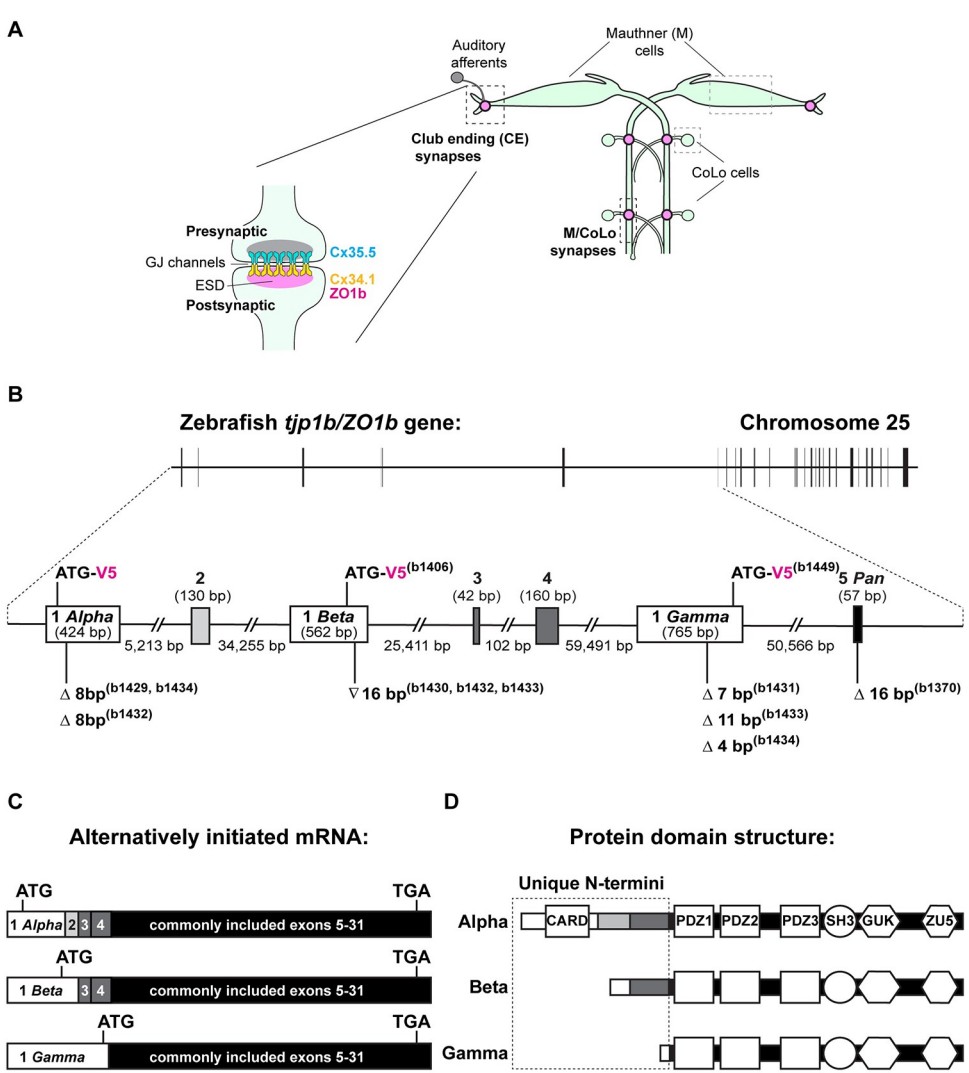

**Fig 1. The *tjp1b/ZO1b* gene is alternatively initiated. A.** Cartoon schematic illustrating the electrical synapses of interest in the Mauthner cell circuit. The image represents a dorsal view with anterior on top. Boxed regions in light grey indicate the Mauthner (M) and CoLo cell bodies, as labeled. Boxed regions in dark grey indicate the stereotypical synaptic contacts used for analysis in this study. Presynaptic auditory afferents contact the postsynaptic Mauthner cell lateral dendrite in the hindbrain forming Club Ending (CE) synapses (denoted by pink circles). In the spinal cord, the presynaptic Mauthner axons form *en passant* electrical synapses with the postsynaptic CoLo interneurons (M/CoLo synapses, denoted by pink circles) in each spinal cord hemisegment (2 of 30 repeating spinal segments are shown). A cartoon enlargement of an electrical synapse in the Mauthner cell circuit appears to the left. Molecularly asymmetric Connexin hemichannels (Cx35.5 [cyan], Cx34.1 [yellow]) directly couple neurons by forming gap junction (GJ) channels. The formation and function of electrical synapses are regulated by ZO1b scaffolds in the electrical synapse density (ESD, magenta). **B.** Schematic diagram of the *tjp1b/ZO1b* gene locus on chromosome 25: 32,031,536–32,271,394 (GRCz11 Ensembl) showing the exon structure located on the forward strand. Horizontal black bar represents the DNA strand and vertical black bars represent individual exons. The region containing alternatively initiated exons for *tjp1b/ZO1b-Alpha*, *-Beta*, and *-Gamma* isoforms is expanded between the dashed lines (Chromosome 25: 32,031,536–32,208,713). Unique initiation exons are represented by white boxes (Exons 1 *Alpha*, 1 *Beta*, and 1 *Gamma*), additional exons are represented by grey boxes, and the first shared exon is represented by a black box (Exon 5). Lengths of exons and introns are indicated. The isoform null mutations generated by CRISPR/ Cas9 for each single and double mutant line (b# designates alleles) are indicated below each Exon 1 (white box). Symbols indicate deletion (Δ) or insertion (∇) of the indicated number of base pairs. The location of the in-frame V5 epitope tag for each isoform is indicated in magenta above each Exon 1. **C.** Schematic diagram representing the alternatively initiated *tjp1b/ZO1b* isoform mRNAs *tjp1b/ZO1b-Alpha* (ENSDART00000173656.2; 7618 bp), *-Beta* (ENSDART00000155992.3; 7626 bp) and *-Gamma* (ENSDART00000112588.5; 7408 bp). Unique 5'UTRs and initiation exons are represented by white boxes, additionally spliced exons are represented by grey boxes and exons shared by all three isoforms are represented by black boxes (Exons 5–31). Approximate location of the ATG start

codon and the shared TGA stop codon are indicated in each transcript. **D.** Schematic diagram of the ZO1b isoform proteins representing ZO1b-Alpha (2005 aa), -Beta (1853 aa), and -Gamma (1689 aa). The unique N-terminal amino acid sequences are indicated in the dashed box. Note that ZO1b-Alpha and ZO1b-Beta both contain amino acid sequence derived from exons 3 and 4. Domains are depicted as white shapes; CARD, PDZ, SH3, GUK and ZU5 are protein-protein interaction modules. White bars represent regions derived from unique initiation exons, grey bars represent regions derived from additionally spliced exons and black bars represent regions derived from shared exons.

broad expression of the *tjp1b/ZO1b* transcript in neural and non-neural cells from 1 to 5 days post fertilization (dpf), suggesting broad usage of the gene in different tissues over time (Fig 2A). This broad expression was expected given ZO1's known roles in tight junction formation in epithelial tissues [15, 16, 22]. While the atlas data suggests expression in developing neurons (Fig 2A), the chemistry used for the scRNA-seq (10X Genomics) is biased toward the 3' end of genes preventing analysis of the alternatively initiated transcripts. Therefore, we designed probes to detect the unique first exons of each isoform and used fluorescent RNA *in situ* on wildtype 5 dpf whole zebrafish. We observed robust expression of *tjp1b/ZO1b-Beta* and *tjp1b/ZO1b-Gamma* transcripts in the brain and spinal cord, yet little evidence for neuronal *tjp1b/ZO1b-Alpha* expression (Fig 2B). We also observed overlapping expression for all isoforms in other areas, such as skin, otolith, and jaw (Fig 2B). From these data we conclude that *tjp1b/ZO1b-Beta* and *tjp1b/ZO1b-Gamma* transcripts are the main isoforms expressed in neuronal tissue.

We next addressed which *tjp1b/ZO1b* isoforms are expressed in neurons of the Mauthner circuit, as the Mauthner neuron and its synapses are an ideal model to understand electrical synapse formation (Fig 1A). This circuit drives a fast escape response to threatening stimuli and is composed of two Mauthner cells in each animal that receive multimodal sensory input that is then relayed to spinal cord circuitry [23–25]. Mauthner makes prominent, stereotyped, electrical synapses with auditory afferents on its lateral dendrite (Club endings, CEs) [26–28]. At CE synapses, the Mauthner lateral dendrite represents the postsynaptic compartment. Additionally, the Mauthner axons make electrical synapses with Commissural Local (CoLo) interneurons of the spinal cord (M/CoLo) [29]. At M/CoLo synapses, the Mauthner axon represents the presynaptic compartment. We looked for *tjp1b/ZO1b* isoform expression in 5 dpf fish harboring the enhancer trap transgene *zf206Et* that expresses GFP in both Mauthner and CoLo cells (Fig 2C–2J). Signals for *tjp1b/ZO1b-Beta* and *tjp1b/ZO1b-Gamma* transcripts were readily observed in both Mauthner and CoLo cell bodies (Fig 2G–2J). We note that much of the fluorescent signal appeared as two distinct puncta within the nuclei of the neurons, similar to other reported RNAs that accumulate into discrete nuclear granules in neuronal cell bodies [30, 31]. Signal for *tjp1b/ZO1b-Alpha* transcript was not detected in Mauthner or CoLo (Fig 2E and 2F) but was expressed at low levels in unidentified nearby cells in the neural tube (white arrow, Fig 2F). We conclude that *tjp1b/ZO1b-Beta* and *tjp1b/ZO1b-Gamma* are expressed in Mauthner and CoLo, making these isoforms candidates for contributing to electrical synapses of this circuit.

## ZO1b-Beta and ZO1b-Gamma localize to electrical synapses

We next determined if distinct transcripts lead to unique protein expression in the Mauthner cell circuit. To achieve this we used CRISPR to insert a V5 epitope tag at the N-terminus of each *tjp1b/ZO1b* isoform and looked for V5 immunostaining in the Mauthner cell circuit of 5 dpf larvae (V5 insertion locations shown in Fig 1B). First, we found that using CRISPR/Cas9-induced homology directed repair with a short, single-stranded nucleotide repair oligo resulted in animals with frequent insertion of the V5 epitope tag into the locus of interest (S1A-S1E Fig). Examining injected animals with mosaically labeled ZO1b isoforms, we found

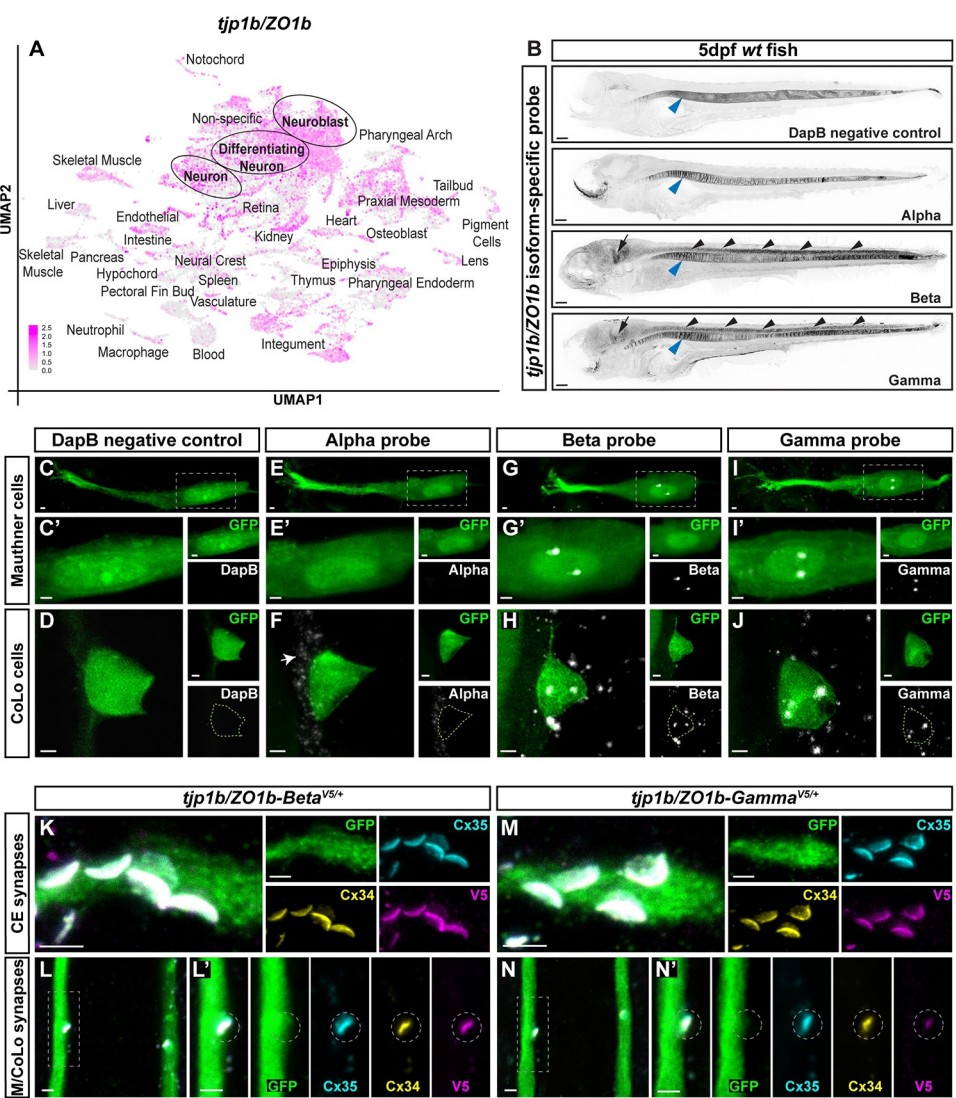

**Fig 2. Alternatively initiated isoforms *tjp1b/ZO1b-Beta* and *tjp1b/ZO1b-Gamma* are expressed in the Mauthner cell circuit and localize to electrical synapses. A.** scRNA-seq dataset of *tjp1b/ZO1b* expression in early development where grey represents low expression and magenta represents the highest level of expression. Cells are derived from whole embryos at 1, 2, and 5 days post fertilization (dpf) and graphed together in UMAP space. Annotated cell types of clusters are indicated and each dot represents a single cell. **B.** Confocal tile scan of fluorescent RNA *in situ* in whole 5 dpf *wt* zebrafish larvae for negative control *DapB*, *tjp1b/ZO1b-Alpha*, *tjp1b/ZO1b-Beta*, and *tjp1b/ZO1b-Gamma*. Images are maximum intensity projections of ~68–98 μm. Anterior left. Contrast is inverted for clarity. Black arrowhead denotes signal in spinal cord. Black arrow denotes signal in hindbrain. While there is possible expression of all three isoforms in the vacuolar cells of the notochord as notochord expression was detected by scRNA-seq methods (Fig 2A), we note that the fluorescent RNA *in situ* method often non-specifically detects signal in the notochord (blue arrowhead). Scale bar = 20 μm. **C-J.** Confocal images of fluorescent RNA *in situ* (white) for the negative control *DapB* (**C, D**), *tjp1b/ZO1b-Alpha* (**E, F**), *tjp1b/ZO1b-Beta* (**G, H**), and *tjp1b/ZO1b-Gamma* (**I, J**) in the Mauthner circuit cell bodies of 5 days post fertilization (dpf) *zf206Et* zebrafish larvae from *wildtype* (*wt*). Animals are co-stained with anti-GFP (green). Scale bars = 2 μm. Images of the Mauthner cell body (C, E, G, I) are maximum intensity projections of ~15–30 μm. Boxed regions are enlarged in C', E', G' and I' and neighboring panels show individual channels. Images of the CoLo cell body (D, F, H, J) are maximum intensity projections of ~6–12 μm and neighboring panels show individual channels. The green dashed line in the probe channel indicates the outline of the CoLo cell body. In F, the arrow indicates the *Alpha* signal in unidentified nearby cells. In G-J, the *Beta* and *Gamma* signals appear as discrete nuclear granules in the neuronal cell body [30, 31]. **K-N.** Confocal images of Mauthner circuit neurons and stereotypical electrical synaptic contacts in 5dpf *zf206Et* zebrafish larvae from heterozygous *V5-tjp1b/ZO1b Beta* (**K, L**) and heterozygous *V5-tjp1b/ZO1b-Gamma* (**M, N**) animals. Animals are stained with anti-GFP (green), anti-Cx35.5 (cyan), anti-Cx34.1 (yellow), and anti-V5 (magenta). Scale bars = 2 μm. Images of the lateral dendrite in the hindbrain (K, M) are maximum intensity projections of ~5–6 μm. Neighboring panels show individual channels. Images of the

sites of contact of M/CoLo processes in the spinal cord (L, N) are maximum-intensity projections of ~3–4 µm. Boxed regions denote stereotyped location of electrical synapses and regions are enlarged in neighboring panels. In L' and N', the white dashed circle denotes the location of the M/CoLo site of contact. Neighboring panels show individual channels. Anterior is up in all images.

that both V5-ZO1b-Beta and V5-ZO1b-Gamma were localized to Mauthner electrical synapses (S1B and S1C Fig). Within the Mauthner circuit, neuronal GJs at both CEs and M/CoLo synapses are made of heterotypic channels formed by Cx35.5, encoded by the gene *gap junction delta 2a* (*gjd2a*), and Cx34.1, encoded by *gjd1a* –both homologous to mammalian Cx36 (*gjd2*) and both recognized by the antibody against human Cx36 [32]. In V5-ZO1b-Beta and V5-ZO1b-Gamma injections, anti-V5 staining was found to colocalize with anti-Cx36 staining at M/CoLo synapses. By contrast, for V5-ZO1b-Alpha, none of the injected embryos mosaically expressed V5-ZO1b-Alpha in any tissue examined including the nervous system, despite 100% of sibling fish being successfully targeted as determined by PCR analysis (S1A Fig). While this suggests that ZO1b Alpha plays no role at Mauthner electrical synapses, it is possible that either no in-frame repairs were made or that V5-ZO1b-Alpha is not expressed at the developmental stage examined. Given the mosaic results, we generated transgenic lines (*V5-tjp1b-Beta^{b1406}* and *V5-tjp1b-Gamma^{b1449}*) and observed localization of V5-ZO1b-Beta and V5-ZO1b-Gamma proteins at the stereotyped locations of CE and M/CoLo electrical synapses (Fig 2K–2N). V5-ZO1b-Beta and V5-ZO1b-Gamma colocalized with neuronal Cx34.1 and Cx35.5 staining at the synaptic contacts and both Beta and Gamma isoforms were distributed with the unique morphological shape of the Mauthner electrical synapses. The localization of both Beta and Gamma isoforms is reminiscent of electrical synaptic ZO1 staining observed using anti-ZO1 antibody [12, 19]. Finally, we also observed V5-ZO1b-Beta and V5-ZO1b-Gamma at non-neuronal epithelial cell junctions (S1D and S1E Fig) and V5-ZO1b-Beta was observed throughout the nervous system colocalized with Connexin staining (S1F Fig). We conclude that both V5-ZO1b-Beta and V5-ZO1b-Gamma are localized to Mauthner electrical synapses.

## ZO1b-Beta is required for robust synaptic Connexin localization

The protein sequences from the three *tjp1b/ZO1b* transcripts predict unique N-terminal peptide sequences for each, though only ZO1b-Alpha has an identifiable domain, a caspase activation and recruitment domain (CARD), while ZO1b-Beta and ZO1b-Gamma have sequences that fail to predict function (Figs 1D and S2A). The common portion of all isoforms retain the PDZ1 binding domain (Fig 1D), which we previously demonstrated mediates direct biochemical interaction with Cx34.1 [12]. We first asked whether differential N-terminal sequences would affect Cx34.1 binding by cloning full-length sequences of *tjp1b/ZO1b-Alpha*, *tjp1b/ZO1b-Beta*, and *tjp1b/ZO1b-Gamma* and using heterologous expression to test for interactions. HEK293T cells were co-transfected with individual ZO1b isoforms and full-length Cx34.1. Using Western blot analysis with antibodies specific to ZO1 and Cx34.1, we found that Cx34.1 was detected in all ZO1b isoform immune complexes compared to control immunoprecipitate (S2B Fig). We conclude that each unique ZO1b isoform can interact with zebrafish Cx34.1.

Next, we examined the role of each ZO1b isoform at electrical synapses *in vivo* using CRISPR/Cas9-induced mutations to knock out individual isoforms. We generated mutant lines that introduced frame-shifts and early stop codons in each of the first coding exons of the isoforms and analyzed antibody staining for Cx35.5, Cx34.1, and ZO1 at Mauthner circuit electrical synapses using immunohistochemistry at 5 dpf (Fig 3A–3J). We compared these

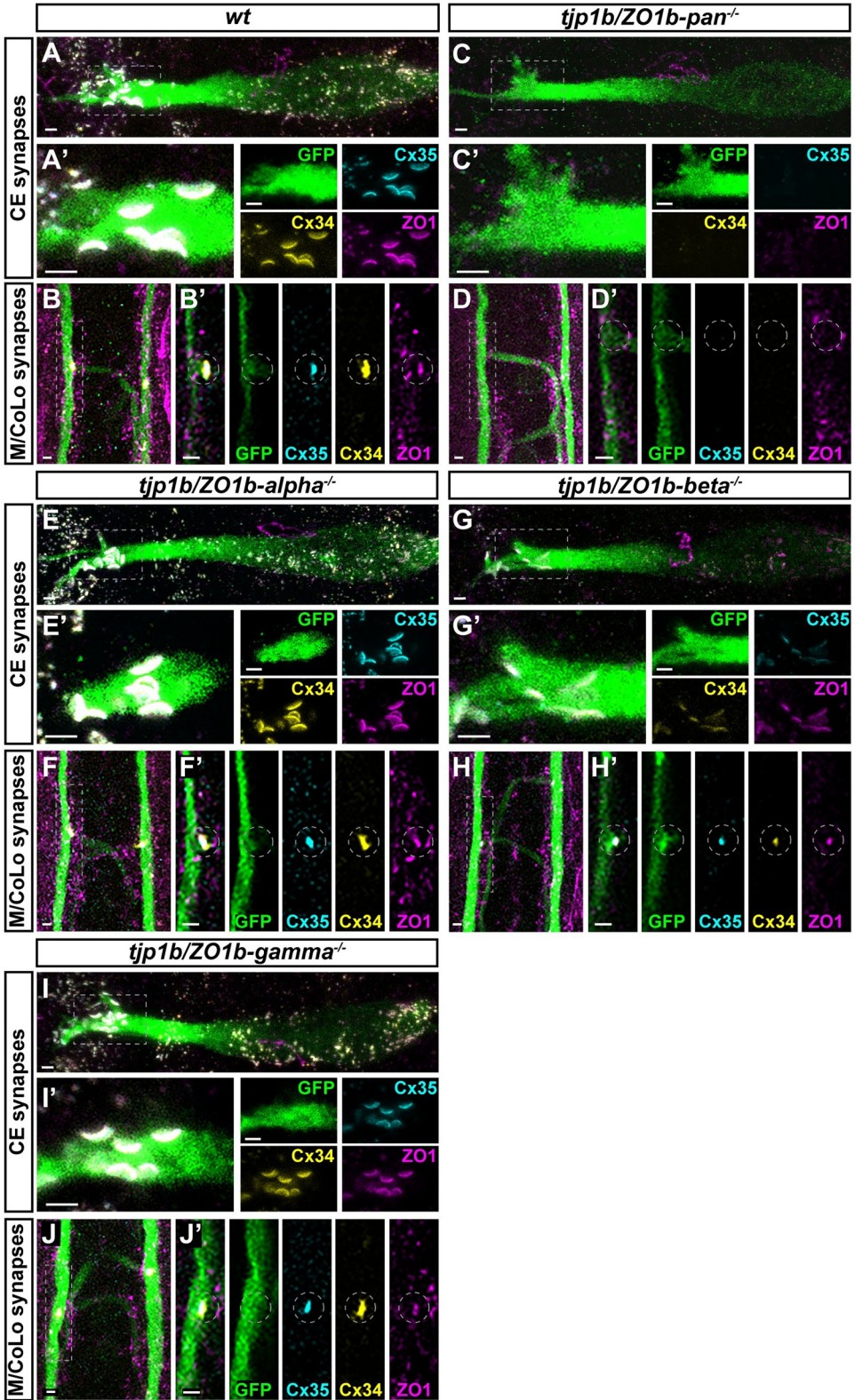

**Fig 3. ZO1b-Beta is necessary for robust Connexin localization to electrical synapses. A-J.** Confocal images of Mauthner circuit neurons and stereotypical electrical synaptic contacts in 5 days post fertilization (dpf) *zf206Et* zebrafish larvae from *wt* (**A, B**), *tjp1b/ZO1b-pan*⁻/⁻ (**C, D**), *tjp1b/ZO1b-alpha*⁻/⁻ (**E, F**), *tjp1b/ZO1b-beta*⁻/⁻ (**G, H**), and

*tjp1b/ZO1b-gamma*<sup>-/-</sup> (**I, J**) animals. Animals are stained with anti-GFP (green), anti-Cx35.5 (cyan), anti-Cx34.1 (yellow), and anti-ZO1 (magenta). Scale bars = 2 μm. Boxed regions denote stereotyped location of electrical synapses and regions are enlarged in neighboring panels. Images of the Mauthner cell body and lateral dendrite in the hindbrain (A, C, E, G, I) are maximum intensity projections of ~10–20 μm. In A', C', E', G', and I', images are maximum-intensity projections of ~3–6 μm and neighboring panels show the individual channels. Images of the sites of contact of M/CoLo processes in the spinal cord (B, D, F, H, J) are maximum-intensity projections of ~6–8 μm. In B', D', F', H' and J', images are from a single 0.42 μm Z-plane and the white dashed circle denotes the location of the M/CoLo site of contact. Neighboring panels show individual channels. Anterior up.

individual-isoform mutants to a "pan" mutation in the first common exon shared by the *alpha*, *beta, and gamma* transcripts (CRISPR mutations associated with particular alleles are identified in Fig 1B). In *tjp1b/ZO1b-pan*<sup>-/-</sup> mutants we observed nearly undetectable Cx35.5, Cx34.1, and ZO1 antibody staining at the CE and M/CoLo electrical synapses (Fig 3C and 3D). As expected from the expression analysis (Fig 2), *tjp1b/ZO1b-alpha*<sup>-/-</sup> mutants had no effect on Connexin or ZO1 staining (Fig 3E and 3F). By contrast, in the *tjp1b/ZO1b-beta*<sup>-/-</sup> mutants there was a reduction of both Connexins and ZO1 staining at Mauthner electrical synapses (Fig 3G and 3H), while the *tjp1b/ZO1b-gamma*<sup>-/-</sup> mutants had little effect on synaptic Connexin staining despite a quantifiable reduction in ZO1 antibody staining (Figs 3I, 3J, and 4). Quantitation of ZO1 at CE synapses and M/CoLo contacts showed similarly diminished staining in both *tjp1b/ZO1b-beta*<sup>-/-</sup> and *tjp1b/ZO1b-gamma*<sup>-/-</sup> isoform mutants. Yet quantitation of Cx34.1 and Cx35.5 at CE synapses and M/CoLo contacts showed greatly diminished synaptic localization in only *tjp1b/ZO1b-beta*<sup>-/-</sup> mutants (Fig 4A and 4B). It is striking that both the Beta and Gamma isoform mutants display a similar decrease in ZO1 antibody staining, yet only the *tjp1b/ZO1b-beta*<sup>-/-</sup> isoform mutants resulted in a concomitant decrease in synaptic Connexin. The reduction in ZO1 staining without an effect on Cx34.1 or Cx35.5 localization in *tjp1b/ZO1b-gamma*<sup>-/-</sup> isoform mutants may reflect an alternative function at electrical synapses independent of Connexin localization. Using qPCR we found that both the *Alpha* and *Gamma* isoforms were upregulated in the *tjp1b/ZO1b-beta*<sup>-/-</sup> mutants (S2C Fig), suggesting that these isoforms might be undergoing transcriptional adaptation [33]. Notably, this shows that the *Alpha* and *Gamma* isoforms were expressed in the *tjp1b/ZO1b-beta*<sup>-/-</sup> mutants but were unable to rescue normal synaptic Connexin localization. We conclude that ZO1b-Beta is required for robust Connexin localization at Mauthner electrical synapses.

## ZO1b-Beta is sufficient for robust Connexin localization to electrical synapses

Next, we examined whether the presence of ZO1b-Beta was sufficient for robust electrical synapse formation. To achieve this we generated double mutants, in cis, amongst each combination of the *tjp1b/ZO1b-alpha*, *-beta*, and *-gamma* isoforms. For double mutants that retained the *tjp1b/ZO1b-beta* isoform we expected to see synaptic Connexin localization, while those lacking the *beta* isoform were expected to show a loss of synaptic Connexin. Indeed, in *tjp1b/ZO1b-alpha-gamma*<sup>-/-</sup> double mutants, in which the Beta isoform remains intact, Cx35.5, Cx34.1 and ZO1 were robustly localized to CE and M/CoLo synapses as compared with wild-type fish (Fig 5A–5D). By contrast, in *tjp1b/ZO1b-alpha-beta*<sup>-/-</sup> and *tjp1b/ZO1b-beta-gamma*<sup>-/-</sup> double mutants, there was decreased signal for Cx35.5, Cx34.1 and ZO1 antibody staining at Mauthner circuit electrical synapses (Fig 5E–5H). Quantitation of Cx35.5, Cx34.1, and ZO1 in both *tjp1b/ZO1b-alpha-beta*<sup>-/-</sup> and *beta-gamma*<sup>-/-</sup> double mutants confirmed the significant loss of synaptic Connexins and ZO1 compared to wildtype at both CE and M/CoLo synapses (Fig 5I and 5J). Quantitation in *tjp1b/ZO1b-alpha-gamma*<sup>-/-</sup> double mutants showed slightly decreased synaptic Cx35.5, Cx34.1, and ZO1 at CE synapses (Fig 5I, CE synapses, Cx35.5:

A

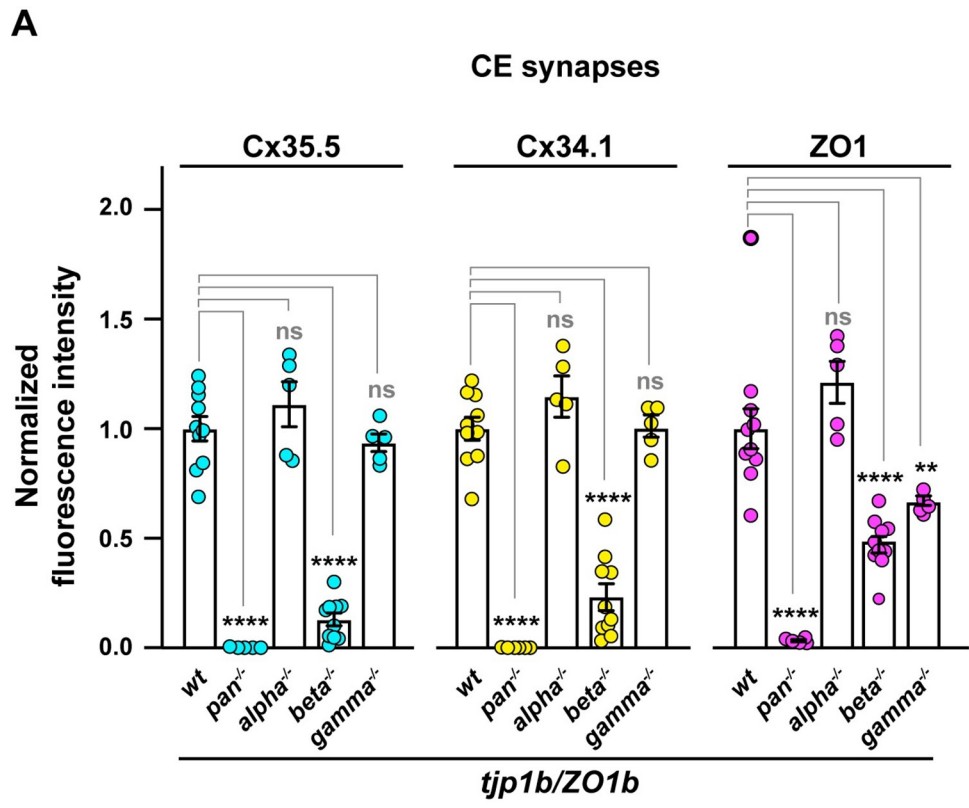

B

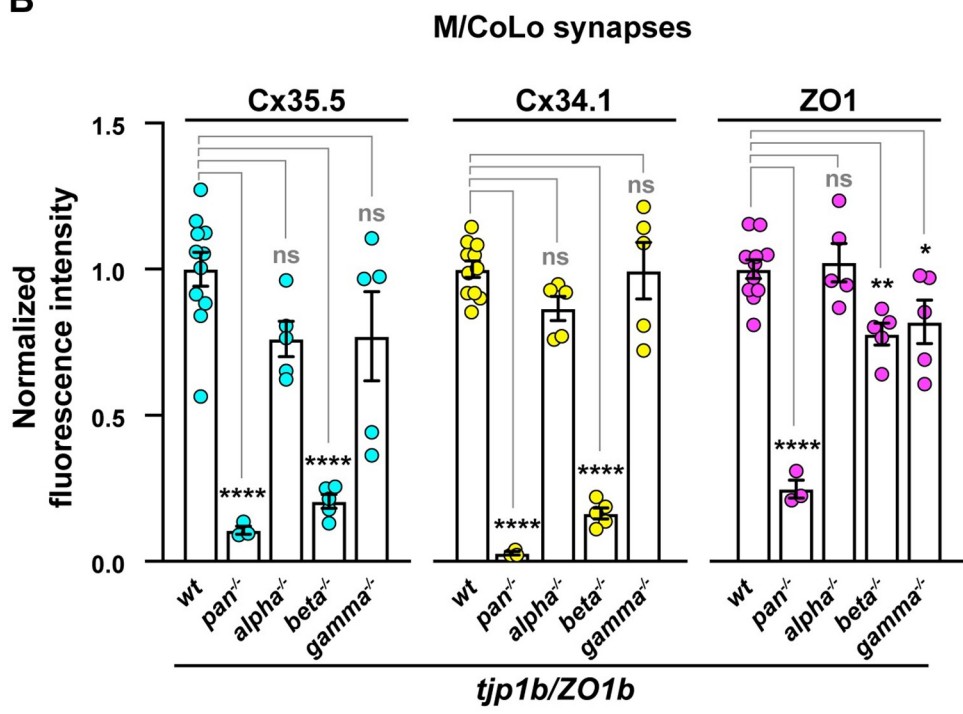

**Fig 4. Quantification of ZO1 isoform mutants. A.** Quantification of Cx35.5, Cx34.1, and ZO1 fluorescence intensities at CE synapses for the noted genotypes. The height of the bar represents the mean of the sampled data normalized to the *wt* average. Circles represent the normalized value of each individual animal. Mean is shown ± SEM.

*wt* n = 10; *tjp1b/ZO1b-pan*$^{-/-}$ n = 5, *tjp1b/ZO1b-alpha*$^{-/-}$ n = 5, *tjp1b/ZO1b-beta*$^{-/-}$ n = 10, and *tjp1b/ZO1b-gamma*$^{-/-}$ n = 5. For Cx35.5 (cyan circles), **** indicates p<0.0001 by ANOVA with Dunnett's test. For Cx34.1 (yellow circles), **** indicates p<0.0001 by ANOVA with Dunnett's test. For ZO1 (magenta circles), **** indicates p<0.0001 and ** indicates p = 0.0093 by ANOVA with Dunnett's test. **B.** Quantification of Cx35.5, Cx34.1, and ZO1 fluorescence intensities at M/CoLo synapses for the noted genotypes. The height of the bar represents the mean of the sampled data normalized to the *wt* average. Circles represent the normalized value of each individual animal. Mean is shown ± SEM. *wt* n = 11, *tjp1b/ZO1b-pan*$^{-/-}$ n = 3, *tjp1b/ZO1b-alpha*$^{-/-}$ n = 5, *tjp1b/ZO1b-beta*$^{-/-}$ n = 5, and *tjp1b/ZO1b-gamma*$^{-/-}$ n = 5. For Cx35.5 (cyan circles), **** indicates p<0.0001 by ANOVA with Dunnett's test. For Cx34.1 (yellow circles), **** indicates p<0.0001 by ANOVA with Dunnett's test. For ZO1 (magenta circles), **** indicates p<0.0001, ** indicates p = 0.008, and * indicates p = 0.0364 by ANOVA with Dunnett's test.

~10% decrease, p = 0.3754, Cx34.1: ~18% decrease, p = 0.0478, ZO1: ~21% decrease, p = 0.0197, ANOVA with Dunnett's test, wildtype n = 15, *tjp1b/ZO1b-alpha-gamma*$^{-/-}$ n = 5), whereas these same proteins slightly increased at M/CoLo synapses (Fig 5J, M/CoLo synapses, Cx35.5: ~43% increase, p = 0.0031, Cx34.1: ~38% increase, p = 0.0013, ZO1: ~16% increase, p = 0.1536, ANOVA with Dunnett's test, wildtype n = 15, *tjp1b/ZO1b-alpha-gamma*$^{-/-}$ n = 5). The opposite effects observed in CE and M/CoLo synapses in *tjp1b/ZO1b-alpha-gamma*$^{-/-}$ double mutants may suggest distinct cell biological mechanisms regulating Connexin and ZO1 localization at each synapse. Yet, the critical observation is that the *tjp1b/ZO1b-alpha-gamma*$^{-/-}$ double mutants retain robust levels of synaptic Connexin and ZO1 localization. From these observations we conclude that the presence of ZO1b-Beta is sufficient for robust Connexin localization at electrical synapses. Taken together, these results provide evidence that both ZO1b-Beta and ZO1b-Gamma are localized to electrical synapses, yet the Beta isoform appears to distinctly license Connexin localization to the synapse during development.

## Discussion

The results presented here reveal that a single genomic locus can encode unique Electrical Synapse Density (ESD) scaffold isoforms with differential effects at the electrical synapse. Our findings show that ZO1b-Beta is the dominant isoform functioning to localize Connexins to Mauthner electrical synapses during development, while ZO1b-Gamma is present but has an alternative, accessory role (Fig 6). We note that both the Beta and Gamma isoforms were also observed to be colocalized with neural Connexins throughout the nervous system, yet disruption of the Beta-isoform, but not the Gamma, resulted in a loss of gross synaptic Connexin localization across the brain (S3 Fig). These findings reveal that protein diversity at the electrical synapse can be encoded by unique isoforms of individual gene loci that have distinct functional impacts on structure.

How the differential function of ZO1b isoforms occurs, with Beta being the dominant scaffold required for developmental Connexin localization, is not currently clear. One possibility is that ZO1b-Beta is present at higher concentration at the electrical synapse, as compared to Gamma, simply providing more scaffold for Connexin binding and localization. While we detected both Beta and Gamma isoforms at electrical synapses using the V5 tag, comparing their relative levels would require unique detection reagents such as alternative epitope tags (e.g., FLAG, HA, etc.), which have not been successful in our experiments to date. If differential expression levels explain the distinct effects on Connexin localization, then expression may be controlled by unique genomic elements upstream of each transcriptional start site (TSS). Currently, there is little known about the transcriptional control of vertebrate electrical synapse formation [34]. As an alternative to protein abundance, the unique amino acids of each isoform may impart distinct functions, perhaps by affecting the functional interactions of ZO1 with Connexins or other ESD proteins at the synapse, or by affecting the sub-synaptic targeting of ZO1. This would be analogous to the effects seen for SAP97 and PSD95 at glutamatergic

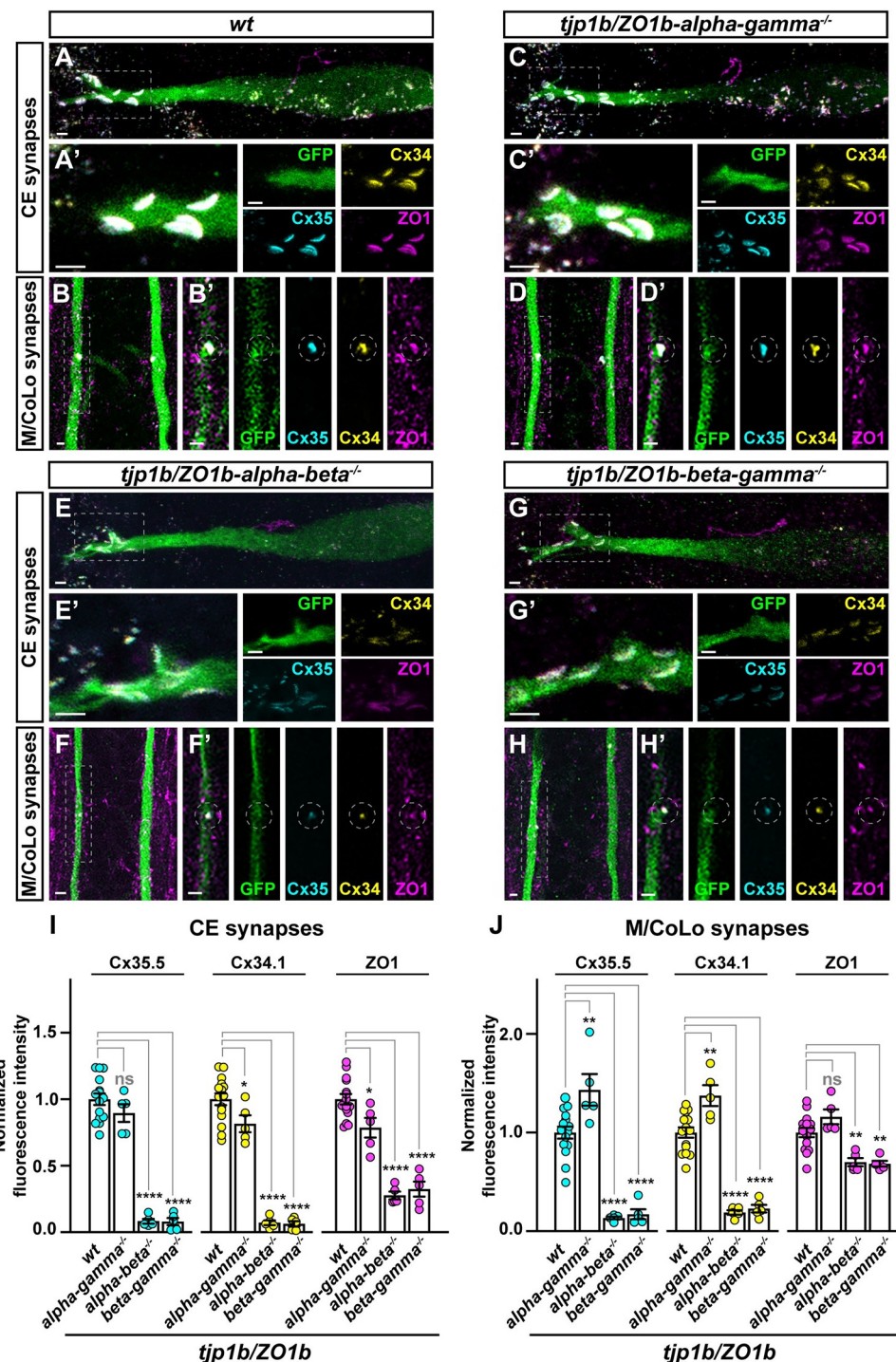

**Fig 5. ZO1b-Beta is sufficient for robust Connexin localization to electrical synapses. A-H.** Confocal images of Mauthner circuit neurons and stereotypical electrical synaptic contacts in 5 days post fertilization (dpf) *zf206Et* zebrafish larvae from *wt* (**A, B**), *tjp1b/ZO1b-alpha-gamma*−/− double mutant (**C, D**), *tjp1b/ZO1b-alpha-beta*−/− double mutant (**E, F**), and *tjp1b/ZO1b-beta-gamma*−/− double mutant (**G, H**) animals. Animals are stained with anti-GFP (green), anti-Cx35.5 (cyan), anti-Cx34.1 (yellow), and anti-ZO1 (magenta). Scale bars = 2 μm. Boxed regions denote stereotyped location of electrical synapses and regions are enlarged in neighboring panels. Images of the Mauthner cell body and lateral dendrite in the hindbrain (A, C, E and G) are maximum intensity projections of ~10–25 μm. In A', C', E' and G', images are maximum-intensity projections of ~3–7 μm and neighboring panels show the individual channels. Images of the sites of contact of M/CoLo processes in the spinal cord (B, D, F and H) are maximum-intensity

projections of ~3–7 μm. In B', D', F' and H', images are from a single 0.42 μm Z-plane and the white dashed circle denotes the location of the M/CoLo site of contact. Neighboring panels show individual channels. Anterior up. **I.** Quantification of Cx35.5, Cx34.1, and ZO1 fluorescence intensities at CE synapses for the noted genotypes. The height of the bar represents the mean of the sampled data normalized to the *wt* average. Circles represent the normalized value of each individual animal. Mean is shown ± SEM. *wt* n = 15, *tjp1b/ZO1b-alpha-gamma*$^{-/-}$ n = 5, *tjp1b/ZO1b-alpha-beta*$^{-/-}$ n = 5, and *tjp1b/ZO1b-beta-gamma*$^{-/-}$ n = 5. For Cx35.5 (cyan circles), **** indicates p<0.0001 by ANOVA with Dunnett's test. For Cx34.1 (yellow circles), **** indicates p<0.0001 and * indicates p = 0.0478 by ANOVA with Dunnett's test. For ZO1 (magenta circles), **** indicates p<0.0001 and * indicates p = 0.0197 by ANOVA with Dunnett's test. **J.** Quantification of Cx34.1 fluorescence intensities at M/CoLo synapses for the noted genotypes. The height of the bar represents the mean of the sampled data normalized to the *wt* average. Circles represent the normalized value of each individual animal. Mean is shown ± SEM. *wt* n = 15, *tjp1b/ZO1b-alpha-gamma*$^{-/-}$ n = 5, *tjp1b/ZO1b-alpha-beta*$^{-/-}$ n = 5, and *tjp1b/ZO1b-beta-gamma*$^{-/-}$ n = 5. For Cx35.5 (cyan circles), ** indicates p = 0.0031 and **** indicates p<0.0001 by ANOVA with Dunnett's test. For Cx34.1 (yellow circles), ** indicates 0.0013 and **** indicates p<0.0001 by ANOVA with Dunnett's test. For ZO1 (magenta circles), *tjp1b/ZO1b-alpha-beta*$^{-/-}$ ** indicates p = 0.0029 and *tjp1b/ZO1b-beta-gamma*$^{-/-}$ ** indicates p = 0.0017 by ANOVA with Dunnett's test.

chemical synapses, where unique scaffold isoforms play distinct functions in synapse development, neurotransmitter receptor localization, and plasticity [13, 14, 35, 36]. By analogy, it is intriguing to speculate that the Beta and Gamma isoforms may play unique roles at the electrical synapse. We have demonstrated that ZO1b-Beta is required for robust Connexin localization during development, and perhaps ZO1b-Gamma could be used to modulate an alternative function such as electrical synapse plasticity, or maturation and homeostasis as the

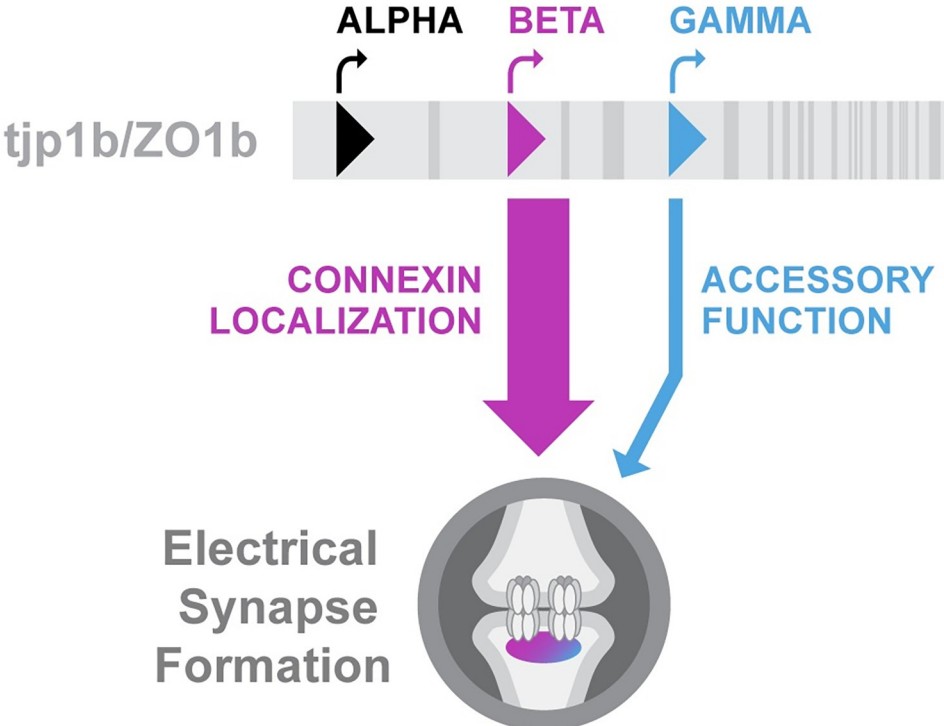

**Fig 6. Model of *tjp1b/ZO1b* isoform contribution to electrical synapse assembly.** Diagram summarizing the contribution of each ZO1b isoform to electrical synapse formation in the Mauthner circuit. Unique first exons of *tjp1b/ZO1b-Alpha* (black), *-Beta* (magenta), and *-Gamma* (blue) are denoted by triangles on the *tjp1b-ZO1b* locus (grey bar) with relative exon structure denoted (dark grey rectangles). Curved arrows indicate unique transcription initiation sites. ZO1b-Beta and -Gamma both localized to the electrical synapse, yet only the ZO1b-Beta isoform is required for robust Connexin (grey ovals) localization to synaptic contacts.

animals grow to adulthood [37–42]. Additionally, we note that our experiments are complicated by the diverse family of ZO-family genes, which in zebrafish includes our gene of focus here, *tjp1b/ZO1b*, and also includes *tjp1a/ZO1a*, *tjp2a/ZO2a*, *tjp2b/ZO2b*, and *tjp3/ZO3*. This family of proteins are all MAGUK scaffolds with related domain structures and, similar to what we have shown here for *tjp1b/ZO1b*, each has complex TSS isoforms. Our experiments cannot exclude the possibility that compensation by these related genes/proteins, or yet to be identified unique TSS isoforms of *tjp1b/ZO1b*, may play a role at the electrical synapse. Indeed, in the mutants we generated, we observed transcriptional adaptation of known *tjp1b/ZO1b* isoforms and this may account for the residual Connexin localization observed in our quantitation. Future experiments that reveal the relative contributions of each isoform, and that probe electrical synapse homeostasis, transmission, and plasticity, will reveal how the isoform-complexity of scaffolds at the electrical synapse regulates structure and function.

An ongoing barrier in the field of electrical synapses is the lack of knowledge surrounding the proteomic makeup of the ESD and how these proteins regulate synapse assembly. Our working model is that ESD complexity comprises multiple unique classes of proteins, including cell adhesion, scaffolds, and regulatory molecules [43, 44]. Our previous work revealed that ZO1b is an exclusively postsynaptic member of the ESD, where it binds Connexins, and is necessary and sufficient for channel localization and function [12]. Our findings here reveal that we must further consider unique isoforms of each of the ESD proteins. Though we focused on ZO1b isoforms resulting from alternative transcriptional initiation, there are many predicted alternatively spliced forms of downstream exons (NCBI) that are likely to alter protein-protein interaction domains–these may add further complexity to building functional electrical synapses. Finally, mammalian ZO1 is known to localize to electrical synapses throughout the brain [45] and the mammalian ZO1 is documented to have multiple isoforms, arising from both unique transcriptional start sites and alternative splicing [16, 17]. This suggests that unique ESD isoforms creating electrical synapse proteomic and functional diversity may be a general feature of vertebrate brains. Thus, ESD complexity, even when viewed only at the level of ZO1b scaffold isoforms, is certainly vastly underestimated. Overall, we expect that ESD complexity will be derived from both discrete classes of molecules as well as unique protein isoforms of the same gene, each with distinct impacts on electrical synapse formation, function, and plasticity.

## Materials and methods

### Ethics statement

Fish were maintained in the University of Oregon fish facility with approval from the Institutional Animal Care and Use Committee (IACUC AUP 21–42).

### Zebrafish

Zebrafish, *Danio rerio*, were bred and maintained at 28°C on a 14 hr on and 10 hr off light cycle. Animals were housed in groups, generally of 25 animals per tank. Development time points were assigned via standard developmental staging [46]. All fish used for this project were maintained in the ABC background developed at the University of Oregon. Most fish had the enhancer trap transgene *zf206Et (M/CoLo:GFP)* in the background [29], unless otherwise noted. The *V5-tjp1b-Beta*[b1406] line contains an in-frame NH$_2$-V5 affinity tag in the exon encoding the *tjp1b/ZO1b-beta* isoform [12]. The *V5-tjp1b-Gamma*[b1449] line contains an in-frame NH$_2$-V5 affinity tag in the exon encoding *tjp1b/ZO1b-gamma* isoform. The *tjp1b-pan*[b1370] mutant line contains a 16 bp deletion in the *tjp1b* gene (common Exon 5, affecting all isoforms) [19]. The *tjp1b-alpha*[b1429] mutant line contains an 8 bp deletion in the *tjp1b/ZO1b-*

*alpha* first exon. The *tjp1b-beta^b1430* mutant line contains a 16 bp insertion in the *tjp1b/ZO1b-beta* first exon. The *tjp1b-gamma^b1431* mutant line contains a 7 bp deletion in the *tjp1b/ZO1b-gamma* first exon. The *tjp1b-alpha-beta^b1432* double mutant line was generated using the *tjp1b-beta^b1430* line and in addition contains an 8 bp deletion in the *tjp1b-alpha* first exon on the same chromosome. The *tjp1b-beta-gamma^b1433* double mutant was generated using the *tjp1b-beta^b1430* line and in addition contains an 11 bp deletion in the *tjp1b-gamma* first exon on the same chromosome. The *tjp1b-alpha-gamma^b1434* double mutant line was generated using the *tjp1b-alpha^b1429* line and in addition contains a 4 bp deletion in the *tjp1b-Gamma* first exon on the same chromosome. Mutant lines were genotyped for all experiments. All immunohistochemistry and RNA *in situ* analyses were performed at 5 dpf. At this stage of development, zebrafish sex is not yet determined [47]. A summary of fish lines generated and used in this study can be found in S4 Fig.

## Cas9-mediated genome engineering of *tjp1b/ZO1b* isoform mutants

Single guide RNAs (sgRNAs) targeting isoform specific Exon 1 of *tjp1b/ZO1b-alpha*, *-beta*, and *-gamma* were designed using the CRISPRscan algorithm [48] and synthesized as previously described [49] using the T7 megascript kit (ThermoFisher).

The CRISPR target sequences used to create mutants for each isoform (PAM underlined) are as follows: *tjp1b Alpha*, 5′-GAGCGGGAGGAGGAATAAAC<u>GGG</u>-3′; *tjp1b Beta*, 5′-G TTTTCTGGGAGGCAGGCTA<u>CGG</u>-3′; and *tjp1b Gamma*, 5′-TGCCCGGGCTGCCTCTAAT A<u>AGG</u> -3′.

Injection mixes were prepared in a pH 7.5 buffer solution of 300 mM KCl and 4 mM HEPES and contained a final concentration of 1–6 uM sgRNA and 10uM Cas9 protein (IDT). Injection mixes were incubated at 37°C for 5 min immediately prior to injection to promote formation of the Cas9 and sgRNA complex. Embryos containing the enhancer trap transgene *zf206Et (M/CoLo:GFP)* were injected with 1 nL at the one-cell stage [50]. Injected embryos were raised to adulthood and outcrossed to wild-type animals. Animals carrying gene-disrupting mutations were identified and verified using PCR and Sanger sequencing. Generating double mutant lines required disrupting each gene on the same chromosome. To generate the isoform double mutant lines, we injected animals carrying known/sequenced mutations in individual isoforms with sgRNAs targeting other isoforms to generate mutations in cis on the same chromosome. For details on lines/lineages, see Zebrafish section above. Confirmation of double mutants in cis were verified by sequencing.

## Cas9-mediated genome engineering of *V5-tjp1b/ZO1b* isoform mosaics and lines

The *V5-tjp1b/ZO1b* isoform-specific single-stranded donor oligos (ssODN) were designed to repair into the respective endogenous *tjp1b/ZO1b* loci. The ssODN contained ~40 bp homology arms which flanked an XbaI restriction site, V5 sequence, and a 5x glycine linker. If the inserted sequence did not disrupt the endogenous sgRNA recognition site, mutations were designed in the CRISPR/PAM sites of the ssODN to prevent further double stranded breaks after repair.

The CRISPR target sequences used to insert N-terminal V5 coding sequence for each isoform (PAM underlined) are as follows:

*tjp1b Alpha*: 5′-ATGCTCTTGTCTGTCTTTGG<u>CGG</u>-3′;

*tjp1b Beta*: 5′-GTATTTCTGGTAATTCACCA<u>TGG</u>-3′;

*tjp1b Gamma*: 5′-TGCCCGGGCTGCCTCTAATA<u>AGG</u>-3′.

The sequences for the ssODN used to repair each *tjp1b* isoform are as follows:

*tjp1b Alpha*: Alt-HDR-Modified-5′-CAAAAATTGCACAGAAATGACCACAAGTCTCTTCT
AAAACATGTCTAGAGGCAAACCGATTCCGAACCCGCTGCTGGGCCTGGATAGCACCGGAGGT
GGAGGCGGACTGCTGTCAGTGTTCGGGGGACCTGTTGAAGAAGTGTTAGCAGGACTCCTCA
AGCTGAAG-3′;

*tjp1b Beta*: 5-TGCGGATTTTTGTGTTTCAAAAAAGCGGTGAACAGTACCATGTCTAGAGGC
AAACCGATTCCGAACCCGCTGCTGGGCCTGGATAGCACCGGAGGTGGAGGCGGAGTGAATTA
CCAGAAATACATTACTGTTATGCAGCTGGCTC-3′;

*tjp1b Gamma*: Alt-HDR-Modified-5′-CTCAACTTGCCTCGGCGCGAGCAGCTAAAGGAAC
AAAATGTCTAGAGGCAAACCGATTCCGAACCCGCTGCTGGGCCTGGATAGCACCGGAGGTGG
AGGCGGATCAGCGCGCGCAGCGTCTAATAAGGTTTGTGCTCATTTATACACATCTCAGACGC
ATTGTCTGCAGTTGTCAG-3′.

Injection mixes were prepared in a pH 7.5 buffer solution of 300 mM KCl and 4 mM HEPES and contained a final concentration of 2uM ssODN, 5–10 uM sgRNA, and 8 uM Cas9 protein (IDT) and were injected as described above. At 3–4 dpf, injected embryos were screened by PCR for integration of the isoform-specific ssODN repair template into the respective *tjp1b* locus using a forward primer against the V5 sequence (5′-CGATTCCGAACCCGCTGCTG-3′) and an isoform-specific reverse primer as follows:

*tjp1b Alpha*: 5′-TCCAAGAAGAGCTCAGCACG-3′;

*tjp1b Beta*: 5′-AACCAAAAATGACACGACAGGCTTTCC-3′;

*tjp1b Gamma*: 5′-ACCGCACATATCGACTAGCG-3′.

At 5 dpf, injected larvae were screened by immunohistochemistry for expression of V5-tjp1b isoforms.

Animals carrying V5-tagged *tjp1b/ZO1b* isoforms were developed by injecting CRISPR reagents and ssODN constructs, raising animals to adulthood, and outcrossing to wild-type animals. Progeny carrying the in-frame V5 modifications were identified by PCR and immunofluorescent staining, and stable lines were established.

## Immunohistochemistry and confocal imaging

Anesthetized, 5 dpf larvae were fixed for 3 hr in 2% trichloroacetic acid in PBS [51]. Fixed tissue was washed in PBS containing 0.5% Triton X-100, followed by standard blocking and antibody incubations. Primary antibody mixes included combinations of the following: rabbit anti-Cx35.5 (Fred Hutch Antibody Technology Facility, clone 12H5, 1:800), mouse IgG2A anti-Cx34.1 (Fred Hutch Antibody Technology Facility, clone 5C10A, 1:350), mouse IgG1 anti-ZO1 (Invitrogen, 33–9100, 1:350), mouse IgG2a anti-V5 (Invitrogen, R960-25, 1:500), rabbit anti-human Cx36 (Invitrogen, 36–4600, 1:500) and chicken anti-GFP (Abcam, ab13970, 1:350–1:500). All secondary antibodies were raised in goat (Invitrogen, conjugated with Alexa-405-Plus, −488, −555, or −633 fluorophores, 1:500). Tissue was then cleared stepwise in a 25%, 50%, 75% glycerol series, dissected, and mounted in ProLong Gold antifade reagent (Thermo-Fisher). Images were acquired on a Leica SP8 Confocal using a 405-diode laser and a white light laser set to 499, 553, 598, and 631 nm, depending on the fluorescent dye imaged. Each laser line's data was collected sequentially using custom detection filters based on the dye. Images of the Club Endings (CEs) were collected using a 63x, 1.40 numerical aperture (NA), oil immersion lens, and images of M/CoLo synapses were collected using a 40x, 1.20 NA, water immersion lens unless otherwise noted in figure legend. For each set of images, the optimal optical section thickness was used as calculated by the Leica software based on the pinhole, emission wavelengths, and NA of the lens. Within each experiment where fluorescence intensity was to be quantified, all animals were stained together using the same antibody mix, processed at the same time, and all confocal settings (laser power, scan speed, gain, offset,

objective, and zoom) were identical. The M/CoLo synapses analyzed were as similar as possible in each fish. The first scan imaged the spinal segments where M/CoLo synapses becomes regularly spaced, which occurs at approximately somite 10 within the spinal cord. The same number of neighboring, caudal somites were imaged and analyzed in all animals. Multiple animals per genotype were analyzed to account for biological variation. To account for technical variation, fluorescence intensity values for each region of each animal were an average across multiple synapses.

## Analysis of confocal imaging

For fluorescence intensity quantitation, confocal images were processed and analyzed using FiJi software [52]. To quantify staining at CE synapses, confocal z-stacks of the Mauthner soma and lateral dendrite were cropped to 200 x 200 pixels centered around the lateral dendritic bifurcation. A FIJIscript cleared the region outside of the Mauthner cell based on GFP staining, and a standard threshold for each channel was set to remove background staining. The image was then transformed into a max intensity projection, synapses thresholded to WT, and the integrated density of each stain within the club ending synapses was measured. To quantify staining at M/CoLo synapses, a standard region of interest (ROI) surrounding each M/CoLo site of contact was drawn and the mean fluorescence intensity was measured. For all experiments, values were normalized to *wildtype* control animals unless otherwise noted, and n represents the number of fish used. Figure images were created using FiJi, Photoshop (Adobe), and Illustrator (Adobe).

## Statistical analysis

Statistical analysis was performed using Prism (GraphPad) software. Standard deviation and standard error of the mean were calculated using Prism. All statistical tests performed and p values are noted in the figure legends. All ANOVA was followed with Dunnett's post-hoc multiple comparison test.

## scRNA-seq analysis

The full methods from the embryo dissociation and cDNA library method can be found from the original dataset publication [20]. In short, whole larvae from the *Tg(olig2*:GFP)vu12 and *Tg (elavl3*:GCaMP6s) backgrounds were pooled (n = 15 per replicate), with 2 replicates at each timepoint (1, 2, 5dpf). Dissociated cells were then run on a 10X Chromium platform using 10x v.2 chemistry aiming for 10,000 cells per run. We aligned reads to the zebrafish genome, GRCz11, using the 10X Cellranger pipeline (version 3.1) using the Lukowicz-Bedford GTF [21] which incorporates genome wide improvements [53]. Transcriptomes were analyzed using the Seurat (V3.1.5) software package for R (V4.1.0) using standard quality control, normalization, and analysis steps. Final UMAP analysis resulted in 49,367 cells with 115 PC dimensions and a resolution of 15.0, which produced 238 clusters. Complete code and data sets can be found at www.adammillerlab.com/resources.

## Fluorescent RNA *in situ*

Custom RNAscope probes to target specific *tjp1b/ZO1b* isoforms were designed and ordered through ACD. For fluorescent *in situs*, we used a modified RNAscope protocol [54]. Briefly, 5 dpf embryos were fixed for 1 h at room temperature in 4% paraformaldehyde / 4% sucrose, rinsed in PBS / 0.1% Tween20, dehydrated through a methanol series and then stored in 100% methanol at −20˚C at least overnight. The methanol was removed and the tissue was air dried

for 30 min at room temperature. The tissue was then exposed to Protease Plus for 1 hr 15 min and washed with PBS / 0.1% Tween 20. The probe was pre-warmed at 40˚C, then cooled. Tissue was incubated with probe overnight at 40˚C. Standard RNAscope V2 multiplex reagents and Opal fluorophores were used, with the modification that 0.2X SSC / 0.1% Tween 20 was used for all wash steps. After *in situ*, tissue was rinsed in PBS / 0.1% Tween 20, then immunostained as described above with the exception that Tween 20 was used as the detergent instead of Triton X-100. Stained tissue was either mounted (whole mount) and imaged in 75% glycerol, or dissected and mounted with ProLong Gold Antifade.

## qPCR analysis

RNA was isolated from 5 dpf larvae using TRIzol reagent, and 5 μg total RNA was used to synthesize cDNA using the Superscript III First strand cDNA synthesis kit (ThermoFisher). qPCR was performed using BioRad CFX Opus 96 Real-Time PCR System thermocycler with the SSsoAdvanced Universal SYBR Green Supermix (BioRad) per manufacturer's instructions. All samples were performed in technical triplicate over three independent experiments. Relative expression of each ZO1b isoform was determined after normalizing to a reference gene using the $2^{-\Delta\Delta CT}$ (Livak) method. Two-tailed unpaired t-test was performed using Prism (GraphPad) to determine statistical significance.

Primers were designed to bind around intron/exon boundaries and to amplify *tjp1b* isoforms. Sequences are as follows:

*tjp1b Alpha*, 5ʹ-TGCTGGAGGAGGTGTGT-3ʹ and 5ʹ-CTGTGCGCTGGTTTAGTTTG-3ʹ (exon 1 *Alpha*—exon 2);

*tjp1b Beta*, 5ʹ-CTGTTATGCAGCTGGCTCTA-3ʹ and 5ʹ-AACTGCAGGGCCGTATTT-3ʹ (exon 1 *Beta*—exon 3);

*tjp1b Gamma*, 5ʹ-CCCGGGCTGCCTCTAATA-3ʹ and 5ʹ-CCTGTGTAGAGTTACAGTGTGC-3ʹ (exon 1 *Gamma*—exon 5);

reference gene *eef1a1l1* (ENSDART00000167847.2), 5ʹ-ATGGCACGGTGACAACAT-3ʹ and 5ʹ-CTCCAGAAGCGTAACACCATT-3ʹ (exon 3 –exon 4).

## Cell culture, transfection, and immunoprecipitation

HEK293T/17 verified cells were purchased from ATCC (CRL-11268; STR profile, amelogenin: X). Cells were expanded and maintained in Dulbecco's Modified Eagle's Medium (DMEM, ATCC) plus 10% fetal bovine serum (FBS, Gibco) at 37˚C in a humidified incubator in the presence of 5% $CO_2$. Low passage aliquots were cryopreserved and stored according to manufacturer's instructions. Cells from each thawed cryovial were monitored for mycoplasma contamination using the Universal Mycoplasma Detection Kit (ATCC, 30–1012K).

*tjp1b-beta* and *-gamma* full-length sequences were cloned from a cDNA library developed from 3 dpf animals using Trizol and SuperScript III, as described above for qPCR. The cDNAs were sequenced to confirm the presence of the major protein-protein interaction domains (Fig 1D), and each clone matched their respective predicted unique N-terminus. *tjp1b-alpha* specific exons 1 and 2 were cloned from genomic DNA, and sequencing revealed two amino acid differences with the predicted unique N-terminus (T78A and E92D, underlined residues in S2A Fig). The full-length cDNA was assembled using Gibson cloning techniques. *tjp1b/ZO1b-alpha* and *tjp1b/ZO1b-beta* were cloned into the pCMV expression vector with an $NH_2$-terminal mVenus tag and a COOH-terminal 8xHIS tag. *tjp1b/ZO1b-gamma* was cloned into the pCMV expression vector with a COOH-terminal 8xHIS tag. Cx34.1 was expressed from a pCMV vector generated in a previous study [12]. Low passage HEK293T/17 cells were seeded 24 hr prior to transfection ($1 \times 10^6$ cells/well of a six-well dish), and the indicated plasmids

were co-transfected using Lipofectamine 3000 (Invitrogen) following the manufacturer's instructions. Cells were collected 36–48 hr post-transfection and lysed in 0.25 ml solubilization buffer (50 mM Tris [pH7.4], 100 mM NaCl, 5 mM EDTA, 1.5 mM MgCl2, 1 mM DTT and 1% Triton X-100) plus a protease inhibitor cocktail (Pierce). Lysates were centrifuged at 20,000 x g for 30 min at 4˚C, and equal amounts of extract were immunoprecipitated with 1.0 µg mouse anti-ZO1 (Invitrogen, 33–9100) overnight with rocking at 4˚C. Immunocomplexes were captured with 25 µl prewashed Protein A/G agarose beads for 1 hr with rocking at 4˚C. Beads were washed three times with lysis buffer, and bound proteins were boiled for 3 min in the presence of LDS-PAGE loading dye containing 200 mM DTT. Samples were resolved by SDS-PAGE using a 4–15% gradient gel and analyzed by Western blot using the following primary antibodies: mouse anti-ZO1 (Invitrogen, 33–9100) and rabbit anti-Cx34.1 3A4 (Fred Hutch Antibody Technology Facility, clone 3A4). Compatible near-infrared secondary antibodies were used for visualization with the Odyssey system (LI-COR).

## Supporting information

**S1 Fig. V5-ZO1b-Beta and V5-ZO1b-Gamma isoform expression. A.** Detection of V5-N-terminal tag integration by CRISPR/Cas9 mediated HDR using a short, single-stranded nucleotide repair oligo. Genomic DNA prepared from individual injected 5 dpf zebrafish was analyzed for successful integration by PCR using a forward primer against the V5 tag and a reverse primer outside the modified region. Products for V5-Alpha (top), V5-Beta (middle) and V5-Gamma (bottom) were resolved by agarose gel electrophoresis (indicated by black arrows). Non-specific products are indicated with an asterisk (*). U = uninjected siblings, NTC = no template control. The table shows the percentage of siblings positive for integration and the percentage of siblings positive for V5 immunostain at any body location in the animal, indicating mosaic expression of V5-tagged ZO1b isoforms. **B,C.** Confocal images of the sites of contact of Mauthner/CoLo processes in the spinal cord of 5 dpf zebrafish larvae mosaically expressing V5-tjp1b/ZO1b-Beta (B) and V5-tjp1b/ZO1b-Gamma (C). V5-tjp1b/ZO1b-Beta animals are stained with anti-Cx35.5 (white), and anti-V5 (magenta). V5-tjp1b/ZO1b-Gamma animals are stained with anti-Cx36 (white), and anti-V5 (magenta). Anterior up. Scale bars = 2 µm. Images are maximum-intensity projections of ~3–4 µm and the dashed circle denotes the M/CoLo site of contact. Neighboring panels show individual channels. **D,E.** Confocal images of spinal cord floor plate collected from heterozygous V5-tjp1b/ZO1b-Beta (**D**) and heterozygous V5-tjp1b/ZO1b-Gamma (**E**) animals. Animals are stained with anti-V5 (magenta). Scale bars = 2 µm. Images are maximum-intensity projections of ~4 µm. Anterior left. **F.** Confocal tile scan of zebrafish brain from 5 dpf *zf206Et* zebrafish larvae from *V5-tjp1b/ZO1b-Beta* animals. Images are maximum intensity projections of ~42 µm. Animals are stained with anti-GFP (green), anti-V5 (magenta), and anti-Cx36 (white). Scale bars = 20 µm. Boxed region denotes stereotyped location of electrical synapses where V5-ZO1b-Beta and Cx36 overlap, and the region is enlarged in neighboring panels. White arrows denote regions where V5-ZO1b-Beta and Cx36 do not overlap. Anterior left. In F', neighboring panels show individual channels.
(TIF)

**S2 Fig. Characterization of ZO1b isoforms. A.** Amino acid sequence alignment of predicted unique N-termini from ZO1b isoforms. Amino acids in grey are unique to each isoform, and amino acids in black are shared between isoforms. Amino acids in black underline indicate the beginning of the sequence common to all isoforms (encoded by exons 5–31). The grey box outlining amino acids in ZO1b-Alpha indicates the CARD domain. Amino acids in grey underline in the CARD domain indicate residues that differ between the published predicted

sequence and the cloned sequence tested in (B). **B.** ZO1b isoform interaction with Cx34.1. HEK293T/17 cells were transfected with plasmids to express Cx34.1 and empty vector (lane 1), ZO1b-Alpha (lane 2), ZO1b-Beta (lane 3), or ZO1b-Gamma (lane 4). Lysates were immunoprecipitated with anti-ZO1 antibody and analyzed by immunoblot for the presence of ZO1b isoform (upper) using anti-ZO1 antibody and Cx34.1 protein using Cx34.1-specific antibody (middle). Total extracts (bottom, 5% input) were blotted for Cx34.1 to demonstrate equivalent expression. Results are representative of three independent experiments. **C.** qPCR analysis of ZO1b isoform mRNA levels in *wt* and *tjp1b/ZO1b-beta*$^{-/-}$ mutants. Relative expression of each ZO1b isoform was determined after normalizing to a reference gene using the $2^{-\Delta\Delta CT}$ (Livak) method. The height of the bar represents the relative fold-expression in *tjp1b/ZO1b-beta*$^{-/-}$ mutants compared to *wt* for each *tjp1b/ZO1b* isoform, as labeled. In *wt*, n = 3 for each tjp1b/ZO1b isoform tested. In *tjp1b/ZO1b-beta*$^{-/-}$ mutants, n = 3 for each tjp1b/ZO1b isoform tested. Circles represent results of three independent experiments. Mean ± SEM is shown. Transcriptional upregulation of all isoforms is observed. In Alpha, * indicates p = 0.0472 by unpaired t-test. In Beta, * indicates p = 0.0234 by unpaired t-test. In Gamma, * indicates p = 0.0232 by unpaired t-test. Note that the *tjp1b/ZO1b-Beta* CRISPR transcript does not undergo nonsense mediated decay.
(TIF)

**S3 Fig. Gross synaptic Connexin localization across the brain. A-C.** Confocal tile scan of zebrafish brain from 5 dpf *zf206Et* zebrafish larvae from the indicated genotypes. Images are maximum intensity projections of ~46 μm. Animals are stained with anti-GFP (green), anti-Cx35.5 (cyan), anti-Cx34.1 (yellow), and ZO1 (magenta). Scale bars = 20 μm. Boxed region denotes stereotyped location of electrical synapses where ZO1b-Beta and Connexins overlap, and the region is enlarged in A'-C' with the neighboring panel showing the Cx34.1 channel. Anterior left.
(TIF)

**S4 Fig. Summary of transgenic fish generated and used in this study. A.** A list of single mutant, double mutant, and V5-affinity tagged animals. Columns detail the *tjp1b/ZO1b* isoform(s) targeted for edit, the exon location and mutation recovered, the background in which the animal was generated, and the line designation assigned.
(TIF)

**S1 Source Data. Fig 4A_Club Ending Synapses.**
(XLSX)

## Acknowledgments

We would like to thank and acknowledge the University of Oregon zebrafish facility staff for superb fish care, especially through the challenges of the global pandemic. We would like to thank the administrative staff at the University of Oregon Institute of Neuroscience.

## Author Contributions

**Conceptualization:** Jennifer Carlisle Michel, Adam C. Miller.

**Data curation:** Jennifer Carlisle Michel, Rachel M. Lukowicz-Bedford, Dylan R. Farnsworth, Adam C. Miller.

**Formal analysis:** Jennifer Carlisle Michel, Rachel M. Lukowicz-Bedford, Dylan R. Farnsworth.

**Funding acquisition:** Adam C. Miller.

**Investigation:** Jennifer Carlisle Michel, Margaret M. B. Grivette, Amber T. Harshfield, Lisa Huynh, Ava P. Komons, Bradley Loomis, Kaitlan McKinnis, Brennen T. Miller, Ethan Q. Nguyen, Tiffany W. Huang, Elias S. Michel, Mia E. Michel, Jane S. Kissinger, Audrey J. Marsh, Lila E. Kaye, Abagael M. Lasseigne, Rachel M. Lukowicz-Bedford, Dylan R. Farnsworth, E. Anne Martin.

**Methodology:** Jennifer Carlisle Michel, Audrey J. Marsh.

**Project administration:** Jennifer Carlisle Michel, Adam C. Miller.

**Resources:** Jennifer Carlisle Michel, Lisa Huynh, Bradley Loomis, Kaitlan McKinnis, Sophia Lauf, Audrey J. Marsh, William E. Crow, Adam C. Miller.

**Supervision:** Jennifer Carlisle Michel, Adam C. Miller.

**Validation:** Jennifer Carlisle Michel, Adam C. Miller.

**Visualization:** Jennifer Carlisle Michel, Rachel M. Lukowicz-Bedford, E. Anne Martin, Adam C. Miller.

**Writing – original draft:** Jennifer Carlisle Michel, Adam C. Miller.

**Writing – review & editing:** Jennifer Carlisle Michel, Adam C. Miller.

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
