## [Decision Letter · Decision Letter 0]

26 Jul 2023

Dear Adam,

Thank you very much for submitting your Research Article entitled 'Electrical synapse formation is differentially regulated by distinct isoforms of a postsynaptic scaffold' to PLOS Genetics.

The manuscript was fully evaluated at the editorial level and by independent peer reviewers. The reviewers appreciated the functional analysis of the distinct ZO1b isoforms in electrical synapse formation but identified some concerns that we ask you to address in a revised manuscript.

We therefore ask you to modify the manuscript according to the review recommendations. Your revisions should address the specific points made by each reviewer.

We hope to receive your revised manuscript within the next 60 days. If you anticipate any delay in its return, we would ask you to let us know the expected resubmission date by email to plosgenetics@plos.org.

Yours sincerely,

Mary

Mary C Mullins

Academic Editor

PLOS Genetics

Gregory Barsh

Editor-in-Chief

PLOS Genetics

Reviewer's Responses to Questions

**Comments to the Authors:**

Reviewer #1: In a previous study, this research group identified the function of Zonula Occludens 1 (ZO1) as a critical scaffolding protein for the recruitment &/or stabilization of Connexin proteins to form functional neuronal gap junctions. In this follow-up study, they use the same larval zebrafish model of electrical synapses in Mauthner Cells to evaluate how three distinct isoforms of ZO1b (-Alpha, -Beta, and -Gamma) influence the localization and accumulation of gap junction proteins Cx34.1 and Cx35.5. They observed expression and protein localization of both -Beta and -Gamma isoforms at Club Ending and Mauthner/CoLo electrical synapses. Using single, double, and triple knockout of ZO1b isoforms, the authors then evaluated the relative contribution of each isoform to the localization of electric synaptic gap junction components. These experiments revealed a predominant role for ZO1b -Beta in localizing Cx34 and Cx35 and support that distinct isoforms of ZO1 play unique roles in the assembly of gap junction components.

Overall, the experiments are rigorous and provide important anatomical evidence that diversity of ZO1 isoforms regulates the assembly of gap junctions at electrical synapses. Minor revisions are recommended regarding the interpretation of the knockout results. While I agree with the conclusion that ZO1b-beta plays a prominent role in electrical synapse formation, the anatomical data alone do not definitively demonstrate that the beta isoform is necessary and sufficient to form electrical synapse. Fig 3 E-H shows smaller, but still formed, electrical synapses in fish lacking the Beta isoform compared to the ZO1b-pan knockout in Fig 2C. Further, the presence of the Beta isoform (Fig 3C) alone leads to the formation of abnormally large synapses, which could also alter synaptic function.

Point by point comments and recommendations are below:

- It is unclear why two major figures discussed in the results are presented as supporting data. The manuscript would be strengthened by making the supporting figures “Fig 1 and Fig 2”, especially since they make up a considerable part of the results section.

- ZO1b-beta -/- (Fig 3G and associated results): While it was shown that the Alpha and Gamma isoforms are upregulated in this mutant, is notable that the accumulation of ZO1 protein appears (at least qualitatively) to be comparably as diminished in accumulation at the postsynapse as the pre- and postsynaptic Connexin proteins. Given that all three isoforms of ZO1b can interact with Cx34.1 (Fig S2B), could the observed difference be that more Beta isoform is able to accumulate &/or be retained at the electrical postsynapse (i.e. it is not that Beta is a better scaffold for Cx34, it’s just that there’s more scaffold for Cx34)?

- It is intriguing that loss of Alpha or Gamma isoform alone does not contribute to changes in Cx34 accumulation (Fig 2 E,I), but that loss of both leads to a significant increase in Cx34. It is possible that the Gamma isoform may be playing an important role in regulating postsynaptic density size, and Alpha may be able to compensate for the loss Gamma (which would be unusual given that Alpha doesn’t normally localize to these synapses)?

- In the interest of open and transparent science, data that is discussed and is not published needs to be shown as a supporting figure (referring to loss of gross synaptic Connexin in the brain in the Beta -/- (but not Gamma -/-) mutants). Alternatively, the authors can remove reference to this observation from the discussion.

- A statistical analysis subsection needs to be added to the Materials and Methods section.

Reviewer #2: This is a beautifully written manuscript supporting the novel claim that distinct protein isoforms of a single gene (ZO1b) make unique contributions to electrical synapse density (ESD). The significance of this work is clear: we know very little about the complex proteome of the ESD; this work demonstrates that the complexity of the ESD is not limited to gene identity, but is further complicated by isoform-specific roles in electrical synapse structure and function. The authors generate a variety of novel mutants and genetic tools specific to each of three ZO1b isoforms. They use these tools to demonstrate that two of the three isoforms are present at stereotyped electrical synapses in the zebrafish, and further show that one isoform, in particular, is essential for localizing connexins to the synapse. The conclusions are well supported by the data.

Major comments:

• T-tests are not appropriate when comparing more than two genotypes. Fig. 2K, Fig. 3I, Fig. S2C should use one-way ANOVAs. This can be easily recalculated in PRISM.

• It would be helpful to have a Table summarizing all the (amazing) mutant and transgenic lines that were used in this study.

• An additional Figure/schematic summarizing the synaptic phenotype of the main mutant isoforms would be helpful (WT/alpha; beta; gamma; beta/gamma/ZO1b-pan) mutants).

• As we only see a single representative image, please quantify the Cx34 label at CE synapses in addition to the M/CoLo synapses in Figures 2/3.

Minor comments/suggestions

• Results: We also observed overlapping expression for “all” isoforms in other areas.

• Results: First section 4th paragraph. It would be helpful to clarify pre/post sites of each synapse here.

• Results. Section 3, 3rd paragraph. While greatly diminished…..mutants retained “residual” Connexin localization…

• Results. Section 3, 3rd paragraph. First sentence. It is unclear that the %’s are being compared to WT (100%).

• In Fig. 1A, the “Club ending (CE) synapses” label is very close to some M/CoLo synapses; this may be confusing for readers not familiar with the system.

• The caption for Fig. 1 is missing a description of panel A (and as such a few panel names are mislabeled).

• Fig. S1 caption, panel A: “Cell[s] are derived…”

• In the first paragraph of the into, I think you meant to say “regulate synaptic vesicle release and neurotransmitter [receptor] localization”.

• Based on the data presented in this manuscript, it is difficult to specifically claim that ZO1b is involved in synapse formation vs. maintenance. It is possible that ZO1b is essential for either recruiting Connexins or stabilizing Connexins at the electrical synapse (or both). Are Connexins present at an earlier stage? Perhaps you could replace “formation is differentially regulated by” with “structure requires distinct” in the title?

• Page and Line numbers would be helpful in any subsequent revisions.

Reviewer #3: Michel 2023 Plos genetics review

This study by Michel et al. provides a clear and concise demonstration of the isoform-specific role of ZO1b in electrical synapse formation in vivo. Fluorescent in situ probes indicate expression of Beta and Gamma ZO1b mRNAs in zebrafish Mauthner and CoLo neurons, but not the Alpha isoform. By inserting a V5 tag into the ZO1b locus at each transcriptional start site, the authors show that protein made from Beta and Gamma colocalizes with pre- and postsynaptic Connexins at electrical synapses in Mauthner and CoLo neurons. All 3 isoforms can interact with Cx34.1 in vitro, but using CRISPR gRNAs to specifically knock out each isoform the authors show that only the Beta isoform is required for normal Cx34/Cx35 expression in vivo. Double-mutant analyses show that the Beta isoform is also sufficient for normal Cx34/Cx35 expression. The design of the study is sound, the manuscript is well-written, and the novel results showing isoform-specific requirements for electrical synapse formation will be of interest to the field. With a few minor revisions the paper’s conclusions can be further strengthened.

Key points to address:

Since this is the first study confirming that multiple isoforms of ZO1b are produced in vivo and not just predicted, it would be important to confirm the full sequence of each expressed isoform. Do they match with the predicted sequences and contain the expected protein domains? The authors likely sequenced each isoform for the HEK cell binding assay, so they could simply clarify that the mRNA/protein structures in Fig. 1C were confirmed by sequencing.

The authors show expression of ZO1b at auditory club ending synapses on Mauthner cell lateral dendrites, but in Fig. 2 and 3 they only quantify expression at the downstream Mauthner/CoLo synapses. Furthermore, they only quantify Cx34.1 intensity. From the images in the figures, it appears that analysis of ZO1 and Cx35 staining would follow the same pattern as for Cx34 at both M/CoLo and club ending synapses. But including these analyses – from images the authors have already collected – would further strengthen the authors’ conclusions, allow for a more complete understanding of how each isoform affects both the pre and postsynaptic side of these electrical synapses, and clarify any potential compensation/adaptation by other ZO proteins.

Along the same lines, the authors mention but say “data not shown” that loss of Beta but not Gamma disrupts Cx expression across the brain. Showing this data would further strengthen the authors’ conclusions that Beta is the key isoform required for Cx expression, not just at M/CoLo synapses, but throughout the CNS.

How do the authors account for the residual Cx labeling in the Alpha-Beta and Beta-Gamma mutants in Fig. 3I? Previously there seemed to be no role for Alpha at these synapses, could ZO1b-Alpha, ZO1a, or ZO2/3 potentially be able to partially compensate for the loss of ZO1b-Beta?

In the Discussion’s first paragraph the authors discuss how differential expression of each isoform may occur. Could mRNA or protein degradation of non-Beta forms also explain or contribute to the observed expression differences between isoforms, either in addition to or instead of proposed but unknown genomic regulatory elements upstream of each start site?

With the caveat that different labeling methods were used, based on the robust V5 labeling of the Gamma isoform (Fig. 1N,O) at club ending and M/CoLo synapses in comparison to the very weak ZO1 staining of Beta mutants (Fig. 2G,H), it appears that Gamma expression may depend on Beta expression. It’s not clear if the construction of the transgenic and mutant lines would allow for this, but analyzing V5-Gamma labeling in Beta mutants vs. siblings could clarify if Gamma expression requires Beta. This could be particularly interesting in light of the fact that Gamma mRNA is upregulated in Beta mutants (Fig S2C), along with the authors’ speculation in the Discussion that each isoform may contribute distinct functions to the electrical synapse, analogous to SAP97 and PSD95 at chemical synapses. Minimally, could the authors discuss what they think the role of Gamma may be in normal fish that exhibit robust expression of both Beta and Gamma at Mauthner electrical synapses?

Minor points to address:

It would be very helpful to clearly indicate the groups being compared in statistical analysis in Fig 2K. Should include comparison of pan vs beta and beta-gamma mutants.

The methods indicate that multiple M/CoLo synapses were analyzed in each fish and averaged, with n representing the number of fish used. Can the authors clarify that the M/CoLo synapses analyzed were always from the same spinal segments in each fish?

Please clarify that fish were imaged and analyzed blind to genotype.

Typo: “noteably” should be “notably” on p. 7, 1st paragraph

**Have all data underlying the figures and results presented in the manuscript been provided?**

Reviewer #1: Yes

Reviewer #2: **No: **Perhaps the data was available but I was not able to access it

Reviewer #3: Yes

PLOS authors have the option to publish the peer review history of their article (what does this mean?). If published, this will include your full peer review and any attached files.

Reviewer #1: No

Reviewer #2: **Yes: **Katie Kindt

Reviewer #3: No

---

## [Decision Letter · Decision Letter 1]

1 Nov 2023

Dear Adam,

We are pleased to inform you that your manuscript entitled "Electrical synapse structure requires distinct isoforms of a postsynaptic scaffold" has been editorially accepted for publication in PLOS Genetics. Congratulations!

Also please address the very minor points made by Reviewer 2 when completing the formatting changes.

Yours sincerely,

Mary

Mary C. Mullins

Academic Editor

PLOS Genetics

Gregory Barsh

Editor-in-Chief

PLOS Genetics

Comments from the reviewers (if applicable):

Reviewer's Responses to Questions

**Comments to the Authors:**

Reviewer #1: All my concerns were sufficiently addressed in this revised manuscript. Beautiful work.

Reviewer #2: The authors have done an admirable job addressing the comments of the reviewers.

only 2 minor comments

Line 288 change to ", while"

It is a good idea to specify what post-hoc multiple comparison test is used for your ANOVAs.

great work.

Reviewer #3: The authors have satisfied all of my concerns, and I congratulate them on a beautiful, well-written, and interesting study.

**Have all data underlying the figures and results presented in the manuscript been provided?**

Reviewer #1: Yes

Reviewer #2: Yes

Reviewer #3: Yes

PLOS authors have the option to publish the peer review history of their article (what does this mean?). If published, this will include your full peer review and any attached files.

Reviewer #1: **Yes: **Lavinia Sheets

Reviewer #2: **Yes: **Katie Kindt

Reviewer #3: No

**Data Deposition**

http://datadryad.org/submit?journalID=pgenetics&manu=PGENETICS-D-23-00652R1

**Press Queries**

---

## [Editor Report · Acceptance letter]

23 Nov 2023

PGENETICS-D-23-00652R1 

Electrical synapse structure requires distinct isoforms of a postsynaptic scaffold 

Dear Dr Miller, 

We are pleased to inform you that your manuscript entitled "Electrical synapse structure requires distinct isoforms of a postsynaptic scaffold" has been formally accepted for publication in PLOS Genetics! Your manuscript is now with our production department and you will be notified of the publication date in due course.

With kind regards,

Anita Estes

PLOS Genetics

On behalf of:
